# Integrated optical frequency division for microwave and mmWave generation

Shuman Sun[1,6], Beichen Wang[1,6], Kaikai Liu[2], Mark W. Harrington[2], Fatemehsadat Tabatabaei[1], Ruxuan Liu[1], Jiawei Wang[2], Samin Hanifi[1], Jesse S. Morgan[1], Mandana Jahanbozorgi[1], Zijiao Yang[1,3], Steven M. Bowers[1], Paul A. Morton[4], Karl D. Nelson[5], Andreas Beling[1], Daniel J. Blumenthal[2✉] & Xu Yi[1,3✉]

The generation of ultra-low-noise microwave and mmWave in miniaturized, chip-based platforms can transform communication, radar and sensing systems[1–3]. Optical frequency division that leverages optical references and optical frequency combs has emerged as a powerful technique to generate microwaves with superior spectral purity than any other approaches[4–7]. Here we demonstrate a miniaturized optical frequency division system that can potentially transfer the approach to a complementary metal-oxide-semiconductor-compatible integrated photonic platform. Phase stability is provided by a large mode volume, planar-waveguide-based optical reference coil cavity[8,9] and is divided down from optical to mmWave frequency by using soliton microcombs generated in a waveguide-coupled microresonator[10–12]. Besides achieving record-low phase noise for integrated photonic mmWave oscillators, these devices can be heterogeneously integrated with semiconductor lasers, amplifiers and photodiodes, holding the potential of large-volume, low-cost manufacturing for fundamental and mass-market applications[13].

Microwave and mmWave with high spectral purity are critical for a wide range of applications[1–3], including metrology, navigation and spectroscopy. Owing to the superior fractional frequency stability of reference-cavity stabilized lasers when compared to electrical oscillators[14], the most stable microwave sources are now achieved in optical systems by using optical frequency division[4–7] (OFD). Essential to the division process is an optical frequency comb[4], which coherently transfers the fractional stability of stable references at optical frequencies to the comb repetition rate at radio frequency. In the frequency division, the phase noise of the output signal is reduced by the square of the division ratio relative to that of the input signal. A phase noise reduction factor as large as 86 dB has been reported[4]. However, so far, the most stable microwaves derived from OFD rely on bulk or fibre-based optical references[4–7], limiting the progress of applications that demand exceedingly low microwave phase noise.

Integrated photonic microwave oscillators have been studied intensively for their potential of miniaturization and mass-volume fabrication. A variety of photonic approaches have been shown to generate stable microwave and/or mmWave signals, such as direct heterodyne detection of a pair of lasers[15], microcavity-based stimulated Brillouin lasers[16,17] and soliton microresonator-based frequency combs[18–23] (microcombs). For solid-state photonic oscillators, the fractional stability is ultimately limited by thermorefractive noise (TRN), which decreases with the increase of cavity mode volume[24]. Large-mode-volume integrated cavities with metre-scale length and a greater than 100 million quality (Q)-factor have been shown recently[8,25] to reduce laser linewidth to Hz-level while maintaining chip footprint

at centimetre-scale[9,26,27]. However, increasing cavity mode volume reduces the effective intracavity nonlinearity strength and increases the turn-on power for Brillouin and Kerr parametric oscillation. This trade-off poses a difficult challenge for an integrated cavity to simultaneously achieve high stability and nonlinear oscillation for microwave generation. For oscillators integrated with photonic circuits, the best phase noise reported at 10 kHz offset frequency is demonstrated in the SiN photonic platform, reaching −109 dBc Hz⁻¹ when the carrier frequency is scaled to 10 GHz (refs. 21,26). This is many orders of magnitude higher than that of the bulk OFD oscillators. An integrated photonic version of OFD can fundamentally resolve this trade-off, as it allows the use of two distinct integrated resonators in OFD for different purposes: a large-mode-volume resonator to provide exceptional fractional stability and a microresonator for the generation of soliton microcombs. Together, they can provide major improvements to the stability of integrated oscillators.

Here, we notably advance the state of the art in photonic microwave and mmWave oscillators by demonstrating integrated chip-scale OFD. Our demonstration is based on complementary metal-oxide-semiconductor-compatible SiN integrated photonic platform[28] and reaches record-low phase noise for integrated photonic-based mmWave oscillator systems. The oscillator derives its stability from a pair of commercial semiconductor lasers that are frequency stabilized to a planar-waveguide-based reference cavity[9] (Fig. 1). The frequency difference of the two reference lasers is then divided down to mmWave with a two-point locking method[29] using an integrated soliton microcomb[10–12]. Whereas stabilizing soliton microcombs to

[1]Department of Electrical and Computer Engineering, University of Virginia, Charlottesville, VA, USA. [2]Department of Electrical and Computer Engineering, University of California Santa Barbara, Santa Barbara, CA, USA. [3]Department of Physics, University of Virginia, Charlottesville, VA, USA. [4]Morton Photonics, Palm Bay, Florida, USA. [5]Honeywell Aerospace Technologies, Plymouth, MN, USA. [6]These authors contributed equally: Shuman Sun, Beichen Wang. ✉e-mail: danb@ucsb.edu; yi@virginia.edu

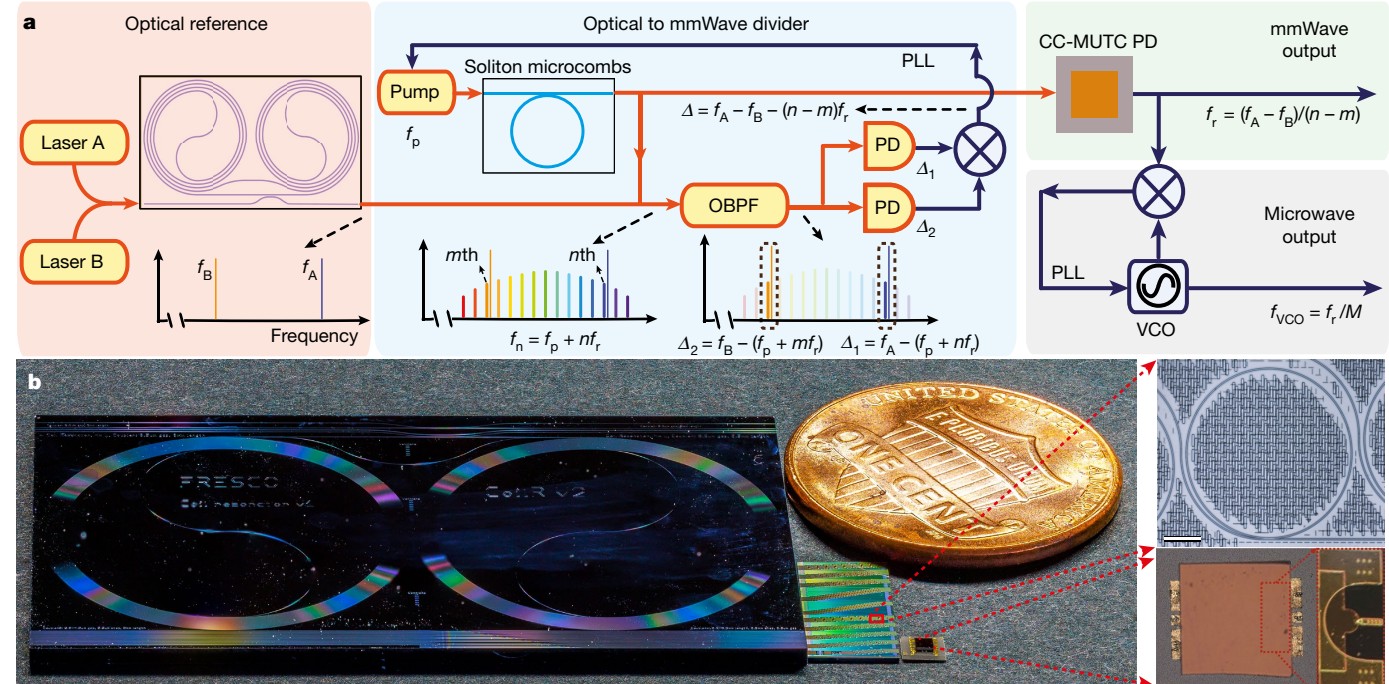

**Fig. 1 | Conceptual illustration of integrated OFD. a**, Simplified schematic. A pair of lasers that are stabilized to an integrated coil reference cavity serve as the optical references and provide phase stability for the mmWave and microwave oscillator. The relative frequency difference of the two reference lasers is then divided down to the repetition rate of a soliton microcomb by feedback control of the frequency of the laser that pumps the soliton. A high-power, low-noise mmWave is generated by photodetecting the OFD soliton microcomb on a CC-MUTC PD. The mmWave can be further divided down to microwave through a mmWave to microwave frequency division with a division ratio of $M$. PLL, phase lock loop. **b**, Photograph of critical elements in the integrated OFD. From left to right are: a SiN 4 m long coil waveguide cavity as an optical reference, a SiN chip with tens of waveguide-coupled ring microresonators to generate soliton microcombs, a flip-chip bonded CC-MUTC PD for mmWave generation and a US 1-cent coin for size comparison. Microscopic pictures of a SiN ring resonator and a CC-MUTC PD are shown on the right. Scale bars, 100 μm (top and bottom left), 50 μm (bottom right).

long-fibre-based optical references has been shown very recently[30,31], its combination with integrated optical references has not been reported. The small dimension of microcavities allows soliton repetition rates to reach mmWave and THz frequencies[12,30,32], which have emerging applications in 5G/6G wireless communications[33], radio astronomy[34] and radar[2]. Low-noise, high-power mmWaves are generated by photo-mixing the OFD soliton microcombs on a high-speed flip-chip bonded charge-compensated modified uni-travelling carrier photodiode (CC-MUTC PD)[12,35]. To address the challenge of phase noise characterization for high-frequency signals, a new mmWave to microwave frequency division (mmFD) method is developed to measure mmWave phase noise electrically while outputting a low-noise auxiliary microwave signal. The generated 100 GHz signal reaches a phase noise of −114 dBc Hz$^{-1}$ at 10 kHz offset frequency (equivalent to −134 dBc Hz$^{-1}$ for 10 GHz carrier frequency), which is more than two orders of magnitude better than previous SiN-based photonic microwave and mmWave oscillators[21,26]. The ultra-low phase noise can be maintained while pushing the mmWave output power to 9 dBm (8 mW), which is only 1 dB below the record for photonic oscillators at 100 GHz (ref. 36). Pictures of chip-based reference cavity, soliton-generating microresonators and CC-MUTC PD are shown in Fig. 1b.

The integrated optical reference in our demonstration is a thin-film SiN 4-metre-long coil cavity[9]. The cavity has a cross-section of 6 μm width × 80 nm height, a free-spectral-range (FSR) of roughly 50 MHz, an intrinsic quality factor of 41 × 10$^6$ (41 × 10$^6$) and a loaded quality factor of 34 × 10$^6$ (31 × 10$^6$) at 1,550 nm (1,600 nm). The coil cavity provides exceptional stability for reference lasers because of its large-mode volume and high-quality factor[9]. Here, two widely tuneable lasers (NewFocus Velocity TLB-6700, referred to as laser A and B) are frequency stabilized to the coil cavity through Pound–Drever–Hall locking technique with a

servo bandwidth of 90 kHz. Their wavelengths can be tuned between 1,550 nm ($f_A$ = 193.4 THz) and 1,600 nm ($f_B$ = 187.4 THz), providing up to 6 THz frequency separation for OFD. The setup schematic is shown in Fig. 2.

The soliton microcomb is generated in an integrated, bus-waveguide-coupled Si$_3$N$_4$ micro-ring resonator[10,12] with a cross-section of 1.55 μm width × 0.8 μm height. The ring resonator has a radius of 228 μm, an FSR of 100 GHz and an intrinsic (loaded) quality factor of 4.3 × 10$^6$ (3.0 × 10$^6$). The pump laser of the ring resonator is derived from the first modulation sideband of an ultra-low-noise semiconductor extended distributed Bragg reflector laser from Morton Photonics[37], and the sideband frequency can be rapidly tuned by a voltage-controlled oscillator (VCO). This allows single soliton generation by implementing rapid frequency sweeping of the pump laser[38], as well as fast servo control of the soliton repetition rate by tuning the VCO[30]. The optical spectrum of the soliton microcombs is shown in Fig. 3a, which has a 3 dB bandwidth of 4.6 THz. The spectra of reference lasers are also plotted in the same figure.

The OFD is implemented with the two-point locking method[29,30]. The two reference lasers are photomixed with the soliton microcomb on two separate photodiodes to create beat notes between the reference lasers and their nearest comb lines. The beat note frequencies are $\Delta_1 = f_A - (f_p + n \times f_r)$ and $\Delta_2 = f_B - (f_p + m \times f_r)$, where $f_r$ is the repetition rate of the soliton, $f_p$ is pump laser frequency and $n$, $m$ are the comb line numbers relative to the pump line number. These two beat notes are then subtracted on an electrical mixer to yield the frequency and phase difference between the optical references and $N$ times of the repetition rate: $\Delta = \Delta_1 - \Delta_2 = (f_A - f_B) - (N \times f_r)$, where $N = n - m$ is the division ratio. Frequency $\Delta$ is then divided by five electronically and phase locked to a low-frequency local oscillator (LO, $f_{LO}$) by feedback

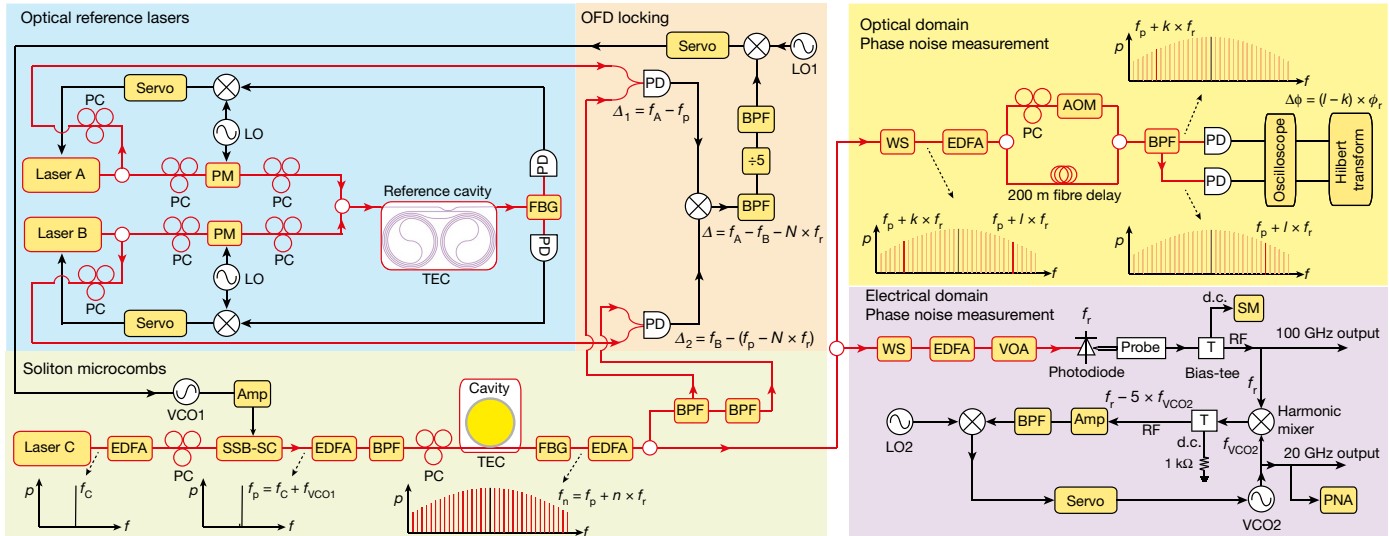

**Fig. 2 | Experimental setup.** A pair of reference lasers is created by stabilizing frequencies of lasers A and B to a SiN coil waveguide reference cavity, which is temperature controlled by a thermoelectric cooler (TEC). Soliton microcomb is generated in an integrated SiN microresonator. The pump laser is the first modulation sideband of a modulated continuous wave laser, and the sideband frequency can be rapidly tuned by a VCO. To implement two-point locking for OFD, the 0th comb line (pump laser) is photomixed with reference laser A, while the $-N$th comb line is photomixed with reference laser B. The two photocurrents are then subtracted on an electrical mixer to yield the phase difference between the reference lasers and $N$ times the soliton repetition rate, which is then used to servo control the soliton repetition rate by controlling the frequency of the pump laser. The phase noise of the reference lasers and the soliton repetition rate can be measured in the optical domain by using dual-tone delayed self-heterodyne interferometry. Low-noise, high-power mmWaves are generated by detecting soliton microcombs on a CC-MUTC PD. To characterize the mmWave phase noise, a mmWave to microwave frequency division is implemented to stabilize a 20 GHz VCO to the 100 GHz mmWave and the phase noise of the VCO can be directly measured by a phase noise analyser (PNA). Erbium-doped fibre amplifiers (EDFAs), polarization controllers (PCs), phase modulators (PMs), single-sideband modulator (SSB-SC), band pass filters (BPFs), fibre-Bragg grating (FBG) filters, line-by-line waveshaper (WS), acoustic-optics modulator (AOM), electrical amplifiers (Amps) and a source meter (SM) are also used in the experiment.

control of the VCO frequency. The tuning of VCO frequency directly tunes the pump laser frequency, which then tunes the soliton repetition rate through Raman self-frequency shift and dispersive wave recoil effects[20]. Within the servo bandwidth, the frequency and phase of the optical references are thus divided down to the soliton repetition rate, as $f_r = (f_A - f_B - 5f_{LO1})/N$. As the local oscillator frequency is in the 10 s MHz range and its phase noise is negligible compared to the optical references, the phase noise of the soliton repetition rate ($S_r$) within the servo locking bandwidth is determined by that of the optical references ($S_o$): $S_r = S_o/N^2$.

To test the OFD, the phase noise of the OFD soliton repetition rate is measured for division ratios of $N = 2, 3, 6, 10, 20, 30$ and $60$. In the measurement, one reference laser is kept at 1,550.1 nm, while the other reference laser is tuned to a wavelength that is $N$ times of the microresonator FSR away from the first reference laser (Fig. 3a). The phase noise of the reference lasers and soliton microcombs are measured in the optical domain by using dual-tone delayed self-heterodyne interferometry[39]. In this method, two lasers at different frequencies can be sent into an unbalanced Mach–Zehnder interferometer with an acoustic-optics modulator in one arm (Fig. 2). Then the two lasers are separated by a fibre-Bragg grating filter and detected on two different photodiodes. The instantaneous frequency and phase fluctuations of these two lasers can be extracted from the photodetector signals by using Hilbert transform. Using this method, the phase noise of the phase difference between the two stabilized reference lasers is measured and shown in Fig. 3b. In this work, the phase noise of the reference lasers does not reach the thermal refractive noise limit of the reference cavity[9] and is likely to be limited by environmental acoustic and mechanical noises. For soliton repetition rate phase noise measurement, a pair of comb lines with comb numbers $l$ and $k$ are selected by a programmable line-by-line waveshaper and sent into the interferometry. The phase noise of their phase differences is measured, and its division by $(l - k)^2$ yields the soliton repetition rate phase noise[39].

The phase noise measurement results are shown in Fig. 3c,d. The best phase noise for soliton repetition rate is achieved with a division ratio of 60 and is presented in Fig. 3c. For comparison, the phase noises of reference lasers and the repetition rate of free-running soliton without OFD are also shown in the figure. Below 100 kHz offset frequency, the phase noise of the OFD soliton is roughly $60^2$, which is 36 dB below that of the reference lasers and matches very well with the projected phase noise for OFD (noise of reference lasers – 36 dB). From roughly 148 kHz (OFD servo bandwidth) to 600 kHz offset frequency, the phase noise of the OFD soliton is dominated by the servo pump of the OFD locking loop. Above 600 kHz offset frequency, the phase noise follows that of the free-running soliton, which is likely to be affected by the noise of the pump laser[20]. Phase noises at 1 and 10 kHz offset frequencies are extracted for all division ratios and are plotted in Fig. 3d. The phase noises follow the $1/N^2$ rule, validating the OFD.

The measured phase noise for the OFD soliton repetition rate is low for a microwave or mmWave oscillator. For comparison, phase noises of Keysight E8257D PSG signal generator (standard model) at 1 and 10 kHz are given in Fig. 3d after scaling the carrier frequency to 100 GHz. At 10 kHz offset frequency, our integrated OFD oscillator achieves a phase noise of −115 dBc Hz⁻¹, which is 20 dB better than a standard PSG signal generator. When comparing to integrated microcomb oscillators that are stabilized to long optical fibres[30], our integrated oscillator matches the phase noise at 10 kHz offset frequency and provides better phase noise below 5 kHz offset frequency (carrier frequency scaled to 100 GHz). We speculate this is because our photonic chip is rigid and small when compared to fibre references and thus is less affected by environmental noises such as vibration and shock. This showcases the capability and potential of integrated photonic oscillators.

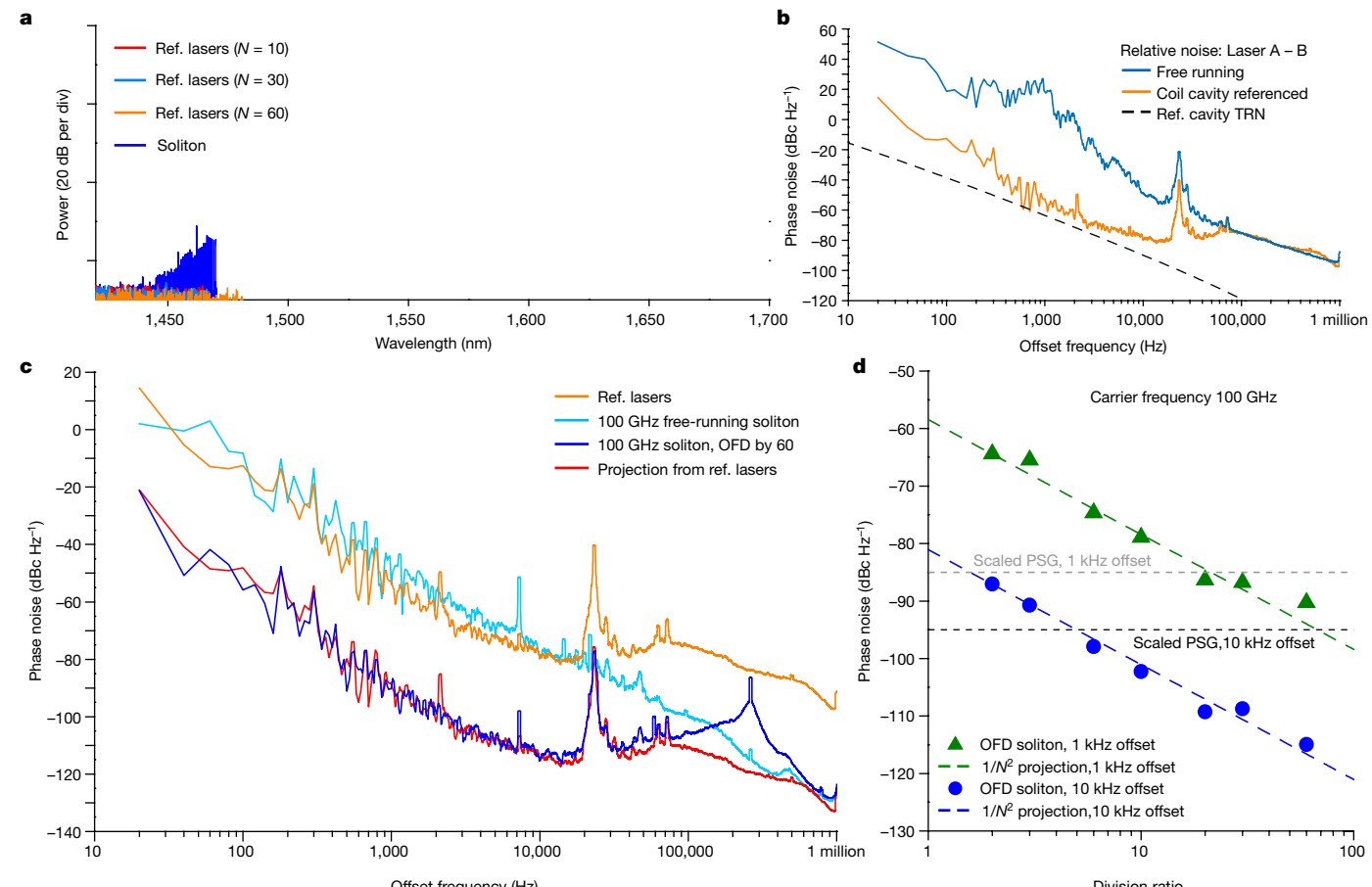

**Fig. 3 | OFD characterization. a**, Optical spectra of soliton microcombs (blue) and reference (Ref.) lasers corresponding to different division ratios. **b**, Phase noise of the frequency difference between the two reference lasers stabilized to coil cavity (orange) and the two lasers at free running (blue). The black dashed line shows the thermal refractive noise (TRN) limit of the reference cavity. **c**, Phase noise of reference lasers (orange), the repetition rate of free-running soliton microcombs (light blue), soliton repetition rate after OFD with a division ratio of 60 (blue) and the projected repetition rate with 60 division ratio (red). **d**, Soliton repetition rate phase noise at 1 and 10 kHz offset frequencies versus OFD division ratio. The projections of OFD are shown with coloured dashed lines.

When comparing to integrated photonic microwave and mmWave oscillators, our oscillator shows exceptional performance: at 10 kHz offset frequency, its phase noise is more than two orders of magnitude better than other demonstrations, including the free-running SiN soliton microcomb oscillators[21,26] and the very recent single-laser OFD[40]. A notable exception is the recent work of Kudelin et al.[41], in which 6 dB better phase noise was achieved by stabilizing a 20 GHz soliton microcomb oscillator to a microfabricated Fabry–Pérot reference cavity.

The OFD soliton microcomb is then sent to a high-power, high-speed flip-chip bonded CC-MUTC PD for mmWave generation. Similar to a uni-travelling carrier PD[42], the carrier transport in the CC-MUTC PD depends primarily on fast electrons that provide high speed and reduce saturation effects due to space-charge screening. Power handling is further enhanced by flip-chip bonding the PD to a gold-plated coplanar waveguide on an aluminium nitride submount for heat sinking[43]. The PD used in this work is an 8-μm-diameter CC-MUTC PD with 0.23 A/W responsivity at 1,550 nm wavelength and a 3 dB bandwidth of 86 GHz. Details of the CC-MUTC PD are described elsewhere[44]. Whereas the power characterization of the generated mmWave is straightforward, phase noise measurement at 100 GHz is not trivial as the frequency exceeds the bandwidth of most phase noise analysers. One approach is to build two identical yet independent oscillators and down-mix the frequency for phase noise measurement. However, this is not feasible for us due to the limitation of laboratory resources. Instead, a new mmWave to microwave frequency division method is developed to coherently divide down the 100 GHz mmWave to 20 GHz microwave, which can then be directly measured on a phase noise analyser (Fig. 4a).

In this mmFD, the generated 100 GHz mmWave and a 19.7 GHz VCO signal are sent to a harmonic radio-frequency (RF) mixer (Pacific mmWave, model number WM/MD4A), which creates higher harmonics of the VCO frequency to mix with the mmWave. The mixer outputs the frequency difference between the mmWave and the fifth harmonics of the VCO frequency: $\Delta f = f_r - 5f_{VCO2}$ and $\Delta f$ is set to be around 1.16 GHz. $\Delta f$ is then phase locked to a stable local oscillator ($f_{LO2}$) by feedback control of the VCO frequency. This stabilizes the frequency and phase of the VCO to that of the mmWave within the servo locking bandwidth, as $f_{VCO2} = (f_r - f_{LO2})/5$. The electrical spectrum and phase noise of the VCO are then measured directly on the phase noise analyser and are presented in Fig. 4b,c. The bandwidth of the mmFD servo loop is 150 kHz. The phase noise of the 19.7 GHz VCO can be scaled back to 100 GHz to represent the upper bound of the mmWave phase noise. For comparison, the phase noise of reference lasers and the OFD soliton repetition rate measured in the optical domain with dual-tone delayed self-heterodyne interferometry method are also plotted. Between 100 Hz to 100 kHz offset frequency, the phase noise of soliton repetition rate and the generated mmWave match very well with each other. This validates the mmFD method and indicates that the phase stability of the soliton repetition rate is well transferred to the mmWave. Below 100 Hz offset frequency, measurements in the optical domain suffer from phase drift

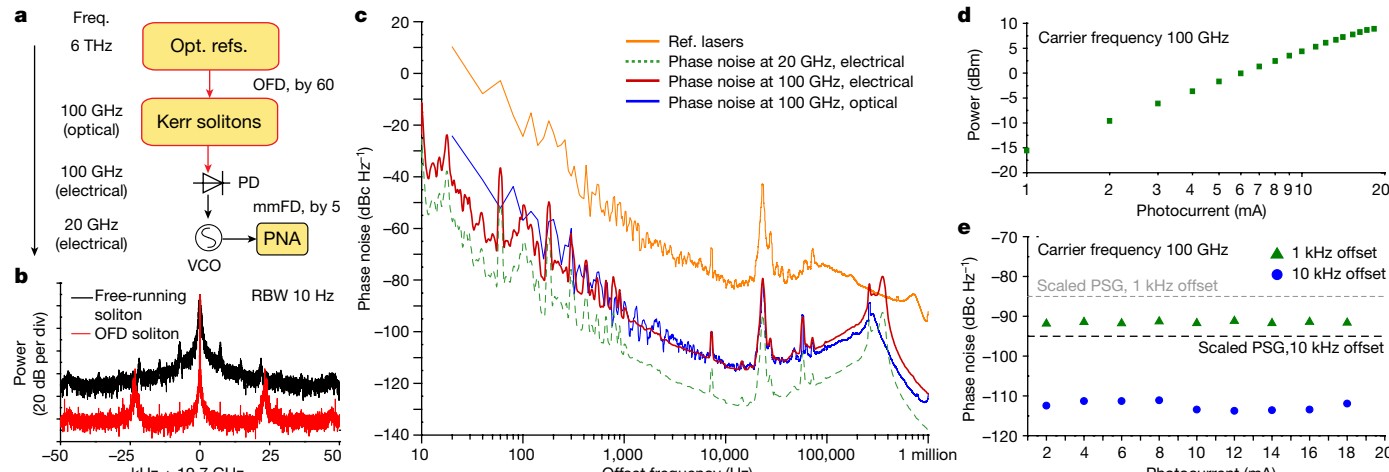

**Fig. 4 | Electrical domain characterization of mmWaves generated from integrated OFD. a**, Simplified schematic of frequency division. The 100 GHz mmWave generated by integrated OFD is further divided down to 20 GHz for phase noise characterization. **b**, Typical electrical spectra of the VCO after mmWave to microwave frequency division. The VCO is phase stabilized to the mmWave generated with the OFD soliton (red) or free-running soliton (black). To compare the two spectra, the peaks of the two traces are aligned in the figure. RBW, resolution bandwidth. **c**, Phase noise measurement in the electrical domain. Phase noise of the VCO after mmFD is directly measured by the phase noise analyser (dashed green). Scaling this trace to a carrier frequency of 100 GHz yields the phase noise upper bound of the 100 GHz mmWave (red). For comparison, phase noises of reference lasers (orange) and the OFD soliton repetition rate (blue) measured in the optical domain are shown. **d**, Measured mmWave power versus PD photocurrent at −2 V bias. A maximum mmWave power of 9 dBm is recorded. **e**, Measured mmWave phase noise at 1 and 10 kHz offset frequencies versus PD photocurrent.

in the 200 m optical fibre in the interferometry and thus yield phase noise higher than that measured with the electrical method.

Finally, the mmWave phase noise and power are measured versus the MUTC PD photocurrent from 1 to 18.3 mA at −2 V bias by varying the illuminating optical power on the PD. Although the mmWave power increases with the photocurrent (Fig. 4d), the phase noise of the mmWave remains almost the same for all different photocurrents (Fig. 4e). This suggests that low phase noise and high power are simultaneously achieved. The achieved power of 9 dBm is one of the highest powers ever reported at 100 GHz frequency for photonic oscillators[36].

## Summary

In summary, we have demonstrated low-noise mmWave and microwave generation by using integrated OFD. The achieved phase noise is primarily limited by the technical noise in the reference lasers and can be improved in the future to the TRN limit of the reference cavity by packaging the reference cavity to isolate environmental noises[9]. Common mode noise cancellation can be further leveraged at that point to reduce the relative noise between the two reference lasers to below the TRN limit. In addition, interesting developments in vacuum-gap cavities using microfabricated mirrors have been shown to overcome the TRN limit of planar-waveguide reference cavities[45]. Chip-based stimulated Brillouin lasers are useful in reducing phase noise of optical references at high offset frequency[17,46]. The division ratio demonstrated in this work is limited by the frequency range of our tuneable lasers instead of the optical span of the soliton microcombs. Microcomb-based OFD that leverages 10s THz optical span or even octave span is possible by using dispersion-engineered microresonators[47]. Furthermore, recent developments in the integration of high-$Q$ resonators and stress-optic modulators[48,49] will enable feedback control of microresonator frequency for OFD, which can greatly increase the OFD servo bandwidth. Therefore, although our demonstration has improved the phase noise of integrated photonic microwave and mmWave oscillators by more than 20 dB, it is feasible to further advance the phase noise by several orders of magnitude in the future, allowing integrated photonic oscillators to, in a certain offset frequency range, for example, 10 kHz,

match some of the best bulk optical OFD oscillators[4]. Finally, there is potential for fully integrated OFD oscillators through heterogeneous integration of SiN reference cavities and microresonators with other components[13], for example, semiconductor lasers, optical amplifiers and photodiodes. In the OFD oscillator, it is possible to replace bulk commercial lasers with integrated lasers for the optical references without compromising the phase noise performance because some newly developed integrated lasers have shown better coherence than the commercial tuneable lasers used in our experiments[50]. Hybrid integration and heterogeneous integration of lasers and soliton microcombs have been shown recently[51,52], and given that the soliton nonlinear conversion efficiency has surpassed 50% with a newly developed pump recycling method[53], it may be possible to eliminate optical amplifiers used in this OFD demonstration in the future.

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

# Article

## Data availability

Data for Figs. 3 and 4 can be accessed at https://doi.org/10.6084/m9.figshare.24813927.

## Code availability

The codes that support the findings of this study are available from the corresponding authors upon reasonable request.

**Acknowledgements** We acknowledge M. Woodson and S. Estrella from Freedom Photonics for MUTC PD fabrication, Ligentec for SiN microresonator fabrication, K. Vahala at Caltech for the access to tuneable optical filters and gratefully acknowledge DARPA GRYPHON (grant no. HR0011-22-2-0008), National Science Foundation (grant no. 2023775), Advanced Research Projects Agency–Energy (grant no. DE-AR0001042). The views and conclusions contained in this document are those of the authors and should not be interpreted as representing official policies of DARPA, ARPA-E or the US Government.

**Author contributions** X.Y., S.S., B.W., Z.Y., D.J.B., S.M.B., A.B. and P.A.M. designed the experiments. S.S., B.W., R.L., F.T. and S.H. performed the system measurements. K.L., M.W.H., J.W., D.J.B., K.D.N. and P.A.M. developed the reference lasers. J.S.M. and A.B. designed and fabricated the CC-MUTC PDs. S.S., B.W., X.Y., F.T., S.M.B. and A.B. analysed the experimental results. All authors participated in preparing the manuscript.

**Competing interests** The authors declare no competing interests.

**Additional information**
**Correspondence and requests for materials** should be addressed to Daniel J. Blumenthal or Xu Yi.
