## [Peer Review File · Nature]

Manuscript Title: Integrated optical frequency division for microwave and mmWave generation

Reviewer Comments & Author Rebuttals

Reviewer Reports on the Initial Version:

Referees' comments:

Referee #1 (Remarks to the Author):

This manuscript discusses significant progress in integrated photonics based generation of low phase noise microwaves. By using an approach termed two point optical frequency division (2P-OFD) and a variety of state-of-the-art integrated photonic components, the group achieves phase noise on a 100 GHz carrier as low as -114 dBc/Hz at 10 KHz frequency offset. This would be equivalent to -134 dBc/Hz at 10 KHz offset if it is scaled down to a more common 10 GHz carrier assuming the standard N^2 scaling law for the phase noise power spectrum. In actual experiments, they build a loop that divides the 100 GHz carrier down to 20 GHz and achieve a phase noise that appears to be about -128 dBc/Hz at 10 KHz offset, which is consistent with the N^2 scaling. These values already show improvement compared to standard electronic frequency generation equipment (text Fig. 4(e) makes a comparison). Although significantly lower phase noise has been demonstrated in frequency division experiments based on mode-locked lasers, here the excitement has to do with the prospects to realize OFD based on highly compact integrated photonics components. This offers potential for synthesis of very low noise microwaves in a much smaller package and with greatly reduced power, which can impact a variety of applications related to timing, navigation, communications and spectroscopy.

This work features several key integrated photonic components, mixed with some discrete components. In particular:

- A thin-film SiN coil cavity with intrinsic Q of $\sim 4 \times 10^7$, a free spectral range of 50 MHz, and an impressive length of 4 meters on chip.
- A pair of commercial, tunable external cavity lasers are locked to different resonances of the same afore-mentioned SiN reference cavity. This allows for large common-mode rejection of the relative phase between the two lasers.
- A soliton microcomb generated via CW pumping of a SiN microresonator. The microresonator has an intrinsic Q of $\sim 4 \times 10^6$ and a free spectral range of 100 GHz. The resulting soliton microcomb has a 3 dB optical bandwidth of 4.6 THz, corresponding to 46 times the 100 GHz spacing of the comb teeth. A different tooth of the microcomb beats with each of the locked external cavity lasers. These beat notes are mixed and then locked to an electronic reference via feedback to the pump laser of the microcomb, which stabilizes the rep rate. The best phase noise is achieved when the two lasers are separated by 60

comb lines. This gives a division factor of 60, leading to approximately 36 dB (60^2) suppression below that of the reference lasers.

- The 100 GHz RF tone is obtained by sending the soliton microcomb onto a CC-MUTC photodiode (charge-compensated modified uni-traveling carrier photodiode) with stated 3 dB bandwidth of 86 GHz. The CC-MUTC detector provides high linearity (low amplitude-to-phase noise conversion), and is capable of high microwave output powers. Here microwave power up to 9 dBm is generated at 100 GHz with 18 mA average photocurrent.

- One of the challenges with soliton microcombs is that the power per comb line is usually low, as is the conversion efficiency of the pump power into the comb. Therefore, in order to have sufficient optical power to generate the photocurrents desired (here 1-18 mA), the optical power is boosted using a cascade of several fiber amplifiers before and after the SiN microresonator where the comb is generated (according to Fig. 2, there are 4 EDFAs between the pump laser and the CC-MUTC photodiode).

The phase noise obtained is better than a Keysight frequency generator by about 20 dB at 10 KHz frequency offset (when the phase noise of the Keysight is scaled up to match the same 100 GHz RF frequency) and is also quite low compared to most other integrated photonics approaches (again when the noise is scaled up to compare the same frequency). One exception, which is noted in the manuscript, refers to a recent preprint (ref. 41, arXiv:2307.08937), which provides ca. 6 dB lower phase noise when comparing at common frequency (20 GHz). Because the current manuscript directly generates 100 GHz - which is on the high side for many practical applications – authors introduce an additional electronic phase locked loop to get down to the more convenient 20 GHz range.

Despite the commendable use of integrated photonic components, the challenges involved in further reducing the footprint and power usage are significant. As one example, the external cavity lasers are still relatively bulky and would need to be replaced by integrated lasers while maintaining good coherence. As another example, the number of optical amplifiers should be minimized, which will be difficult given the poor power efficiencies of the soliton microcombs. Some of the possibilities for further performance improvement and SWAP reduction are discussed in the Summary section, but it is clear that despite the very nice performance demonstrated here, there is still a long road to realize OFD based on highly compact integrated photonics components in a footprint well below that of mode-locked laser based systems.

Referee #2 (Remarks to the Author):

The authors demonstrate low noise mmWave and microwave generation by using integrated optical frequency division. They combine a range of established techniques, including two-point optical frequency division, microresonator soliton frequency comb, and high-power and high-speed modified uni-traveling carrier photodetectors. The results are solid. I agree that the crucial photonic components used here hold promise for potential heterogeneous integration onto a single chip. However, the

demonstrated system appears to be relatively bulky, and the noise performance does not yet match that of commercial products with standard packaging, such as X-LNO from QuantX, and hQp Photonic microwave oscillator. Additionally, the underlying idea is well-established within the field and the manuscript does not introduce a new or groundbreaking concept. Given these considerations, the novelty and significance of this work may be somewhat limited for publication in a prestigious journal like Nature.

Specific comments are below

- 1) The phase noise of the demonstrated microwave signal is noticeably suppressed at low frequency offsets within the locking bandwidth. This reduction is achieved through referencing to the Si₃N₄ coil reference cavity. However, at higher offset frequencies, the dominant noise source stems from the soliton frequency comb. It would be interesting to evaluate the noise performance across a wider frequency range to gain a comprehensive understanding of the system's behavior.
- 2) It would be valuable to determine the exact locking bandwidths for the various frequency servo loops in the system, including the locking of reference lasers A and B to the coil reference cavity, the locking of the microcomb to the reference lasers, and the mmFD servo loop. These bandwidths play a crucial role in understanding the system's performance and limitations.
- 3) On page 6, the authors claim "Above 100 kHz offset frequency, the mmFD measured phase noise is limited by the strong mmFD servo locking bump". This claim is not clear to me since the microcomb locking introduces a big bump around 250 kHz from Fig.3(c).
- 4) It is mentioned that an additional LO2 is required for mmFD. Could the authors elaborate on the specific purpose or necessity of incorporating LO2 into the mmFD system?
- 5) On page 3, $f_r = (f_A - f_B - f_{LO1})/N$. Here, f_{LO1} should be Δ .

Referee #3 (Remarks to the Author):

In their manuscript on "Integrated optical frequency division for stable microwave and mmWave generation" the authors devise and investigate a technology that allows very good phase noise. The scheme requires commercial ECDL lasers that are stabilized by being locked to a 4 metre long integrated coil cavity. Both of them are then used to stabilize a narrow optical frequency comb with the two point optical frequency division. The resulting phase noise stability is very impressive for an all compact and integratable device. It is still, however, short of the stability reached by fibre-optic/mode lock laser based OFD. In order to quantify the phase noise of the high frequency 100GHz mmWave the authors furthermore develop a new technique that is very interesting.

The work is original and the results significant and well reference, the conclusions are robust.

The achieved power of 9 dBm at 100 GHz is impressive. What would be required to achieve even higher values? What would be required to use this system to go to even higher frequencies up to 300 GHz? Difficult to measure, but powerful light sources would be very sought after.

Author Rebuttals to Initial Comments:

Referees' comments:

Referee #1 (Remarks to the Author):

This manuscript discusses significant progress in integrated photonics based generation of low phase noise microwaves. By using an approach termed two point optical frequency division (2P-OFD) and a variety of state-of-the-art integrated photonic components, the group achieves phase noise on a 100 GHz carrier as low as -114 dBc/Hz at 10 KHz frequency offset. This would be equivalent to -134 dBc/Hz at 10 KHz offset if it is scaled down to a more common 10 GHz carrier assuming the standard N^2 scaling law for the phase noise power spectrum. In actual experiments, they build a loop that divides the 100 GHz carrier down to 20 GHz and achieve a phase noise that appears to be about -128 dBc/Hz at 10 KHz offset, which is consistent with the N^2 scaling. These values already show improvement compared to standard electronic frequency generation equipment (text Fig. 4(e) makes a comparison). Although significantly lower phase noise has been demonstrated in frequency division experiments based on mode-locked lasers, here the excitement has to do with the prospects to realize OFD based on highly compact integrated photonics components. This offers potential for synthesis of very low noise microwaves in a much smaller package and with greatly reduced power, which can impact a variety of applications related to timing, navigation, communications and spectroscopy.

This work features several key integrated photonic components, mixed with some discrete components. In particular:

- A thin-film SiN coil cavity with intrinsic Q of $\sim 4 \times 10^7$, a free spectral range of 50 MHz, and an impressive length of 4 meters on chip.*
- A pair of commercial, tunable external cavity lasers are locked to different resonances of the same aforementioned SiN reference cavity. This allows for large common-mode rejection of the relative phase between the two lasers.*
- A soliton microcomb generated via CW pumping of a SiN microresonator. The microresonator has an intrinsic Q of $\sim 4 \times 10^6$ and a free spectral range of 100 GHz. The resulting soliton microcomb has a 3 dB optical bandwidth of 4.6 THz, corresponding to 46 times the 100 GHz spacing of the comb teeth. A different tooth of the microcomb beats with each of the locked external cavity lasers. These beat notes are mixed and then locked to an electronic reference via feedback to the pump laser of the microcomb, which stabilizes the rep rate. The best phase noise is achieved when the two lasers are separated by 60 comb lines. This gives a division factor of 60, leading to approximately 36 dB (60^2) suppression below that of the reference lasers.*
- The 100 GHz RF tone is obtained by sending the soliton microcomb onto a CC-MUTC photodiode (charge-compensated modified uni-traveling carrier photodiode) with stated 3 dB bandwidth of 86 GHz. The CC-MUTC detector provides high linearity (low amplitude-to-phase noise conversion), and is capable of high microwave output powers. Here microwave power up to 9 dBm is generated at 100 GHz with 18 mA average photocurrent.*
- One of the challenges with soliton microcombs is that the power per comb line is usually low, as is the conversion efficiency of the pump power into the comb. Therefore, in order to have sufficient optical power to generate the photocurrents desired (here 1-18 mA), the optical power is boosted using a cascade of several fiber amplifiers before and after the SiN microresonator where the comb is generated (according to Fig. 2, there are 4 EDFAs between the pump laser and the CC-MUTC photodiode).*

The phase noise obtained is better than a Keysight frequency generator by about 20 dB at 10 KHz frequency offset (when the phase noise of the Keysight is scaled up to match the same 100 GHz RF frequency) and is also quite low compared to most other integrated photonics approaches (again when the noise is scaled up to compare the same frequency). One exception, which is noted in the manuscript, refers to a recent preprint (ref. 41, arXiv:2307.08937), which provides ca. 6 dB lower phase noise when comparing at common frequency (20 GHz). Because the current manuscript directly generates 100 GHz - which is on the high side for many practical applications – authors introduce an additional electronic phase locked loop to get down to the more convenient 20 GHz range.

Despite the commendable use of integrated photonic components, the challenges involved in further reducing the footprint and power usage are significant. As one example, the external cavity lasers are still relatively bulky and would need to be replaced by integrated lasers while maintaining good coherence. As another example, the number of optical amplifiers should be minimized, which will be difficult given the poor power efficiencies of the soliton microcombs. Some of the possibilities for further performance improvement and SWAP reduction are discussed in the Summary section, but it is clear that despite the very nice performance demonstrated here, there is still a long road to realize OFD based on highly compact integrated photonics components in a footprint well below that of mode-locked laser based systems.

Reply:

We thank the reviewer for his/her detailed summary of the highlights and challenges of our demonstration in this manuscript.

Indeed, in our demonstration, the key components that provide phase stability and optical frequency division are based on integrated photonics, including optical reference cavity, optical frequency combs, and photodiodes. This validates that the optical frequency division and low phase noise can be achieved with integrated photonics. While the demonstrated system includes components that are not compact, e.g., external cavity lasers and EDFAs, we hope our demonstration illustrates that integrated optical frequency division is a promising approach towards miniaturized, ultra-stable microwave/mmWave oscillators. In the revised manuscript, we added more discussions regarding some very recent developments of both integrated lasers and microcomb efficiency that are highly relevant to fully miniaturizing the OFD system.

In the revised manuscript, we added the following sentences in the summary: “...*In the OFD oscillator, it is possible to replace bulk commercial lasers with integrated lasers for the optical references without compromising the phase noise performance since some newly developed integrated lasers have shown better coherence than the commercial tunable lasers used in our experiments [51]. Hybrid integration and heterogeneous integration of lasers and soliton microcombs have been shown recently [52,53], and given that the soliton nonlinear conversion efficiency has surpassed 50% with a newly developed pump recycling method [54], it may be possible to eliminate optical amplifiers used in this OFD demonstration in the future.*”

Referee #2 (Remarks to the Author):

The authors demonstrate low noise mmWave and microwave generation by using integrated optical frequency division. They combine a range of established techniques, including two-point optical frequency division, microresonator soliton frequency comb, and high-power and high-speed modified uni-traveling carrier photodetectors. The results are solid. I agree that the crucial photonic components used here hold promise for potential heterogeneous integration onto a single chip. However, the demonstrated system appears to be relatively bulky, and the noise performance does not yet match that of commercial products with standard packaging, such as X-LNO from QuantX, and hQp Photonic microwave oscillator. Additionally, the underlying idea is well-established within the field and the manuscript does not introduce a new or groundbreaking concept. Given these considerations, the novelty and significance of this work may be somewhat limited for publication in a prestigious journal like Nature.

Reply: We thank the reviewer for his/her insightful comments. Our demonstration is the first attempts to realize the integrated version of optical frequency division (OFD), as the phase stability and optical frequency division are all based on integrated components, including optical reference cavity, optical frequency combs, and photodiodes. The main novelty and significance of the work is to show that optical frequency division and low phase noise can be achieved with integrated photonics, which paves a promising path towards miniaturized, fully integrated, ultra-stable microwave/mmWave oscillators. In terms of performance, our demonstration, while improving the phase noise performance of integrated photonic microwave/mmWave oscillators by more than 100 times, does not represent the limit for integrated photonic oscillators. A number of follow-up developments can be done to further improve the phase noise, including using a larger frequency division ratio, higher servo locking bandwidth for reference lasers and optical frequency division, and packaging to isolate environmental noise from coupling into the oscillator. While some of these technical developments might be challenging, we hope our demonstration has shown that it is possible for integrated photonic OFD oscillators to approach, or even exceed, the phase noise levels that are previously exclusive to non-chip-based oscillators, such as bulk optics OFD, microwave sapphire resonators (X-LNO from QuantX), and oscillators referenced to optical fibers (hQp Photonic microwave oscillators). Finally, when compared with other methods, integrated OFD oscillators have the advantage of conveniently reaching mmWave and sub-THz frequency bands for a wide range of applications.

Specific comments are below

1) The phase noise of the demonstrated microwave signal is noticeably suppressed at low frequency offsets within the locking bandwidth. This reduction is achieved through referencing to the Si₃N₄ coil reference cavity. However, at higher offset frequencies, the dominant noise source stems from the soliton frequency comb. It would be interesting to evaluate the noise performance across a wider frequency range to gain a comprehensive understanding of the system's behavior.

Reply: We thank the reviewer for this valuable suggestion. Indeed, at higher offset frequency, beyond the OFD servo bandwidth, the phase noise of the repetition rate no longer follows the optical references. From 148 kHz (OFD servo bandwidth) to around 600 kHz, the phase noise is dominated by the servo pump of the OFD locking loop. Above 600 kHz, the phase noise follows the phase noise of the free-running soliton, which is likely to be affected by the noise of the pump laser [ref.14].

We have included this on page 5 in the revised manuscript: “... From roughly 148 kHz (OFD servo bandwidth) to 600 kHz offset frequency, the phase noise of OFD soliton is dominated by the servo pump of the OFD locking loop. Above 600 kHz offset frequency, the phase noise follows that of the free-running soliton, which is likely to be affected by the noise of the pump laser [14].”

2) It would be valuable to determine the exact locking bandwidths for the various frequency servo loops in the system, including the locking of reference lasers A and B to the coil reference cavity, the locking of the microcomb to the reference lasers, and the mmFD servo loop. These bandwidths play a crucial role in understanding the system's performance and limitations.

Reply: We thank the reviewer for this helpful comment. Our original measurements contain information regarding locking bandwidths. We estimate the locking bandwidth by the offset frequency where the noise of the locked oscillator increases to that of the same oscillator at the free-running condition (equivalent to 0 dB gain). For the locking of reference lasers A and B to the coil reference cavity, the locking bandwidth is 90 kHz (Fig.3b in the manuscript). For the locking of the microcomb to the reference lasers, the locking bandwidth is around 148 kHz (Fig. 3c in the manuscript). For the mmFD servo loop, the locking bandwidth is 150 kHz (see figure below).

We have included this information in the revised manuscript.

Figure 1 Phase noise measurement of VCO2 when it is free running (blue) and when its 5th harmonics is servo locked to the 100 GHz mmWave in the mmFD (green). The carrier frequency of the VCO2 is 19.7 GHz.

3) On page 6, the authors claim “Above 100 kHz offset frequency, the mmFD measured phase noise is limited by the strong mmFD servo locking bump”. This claim is not clear to me since the microcomb locking introduces a big bump around 250 kHz from Fig.3(c).

Reply: We thank the reviewer for pointing this out. This statement is indeed confusing, and we have removed it in the revised manuscript. In the revised manuscript, this information is replaced by the 150 kHz mmFD servo bandwidth, which is a more accurate description of the mmFD servo loop.

We have added this sentence on page 6 in the revised manuscript: “The bandwidth of the mmFD servo loop is 150 kHz.”

4) It is mentioned that an additional LO2 is required for mmFD. Could the authors elaborate on the specific purpose or necessity of incorporating LO2 into the mmFD system?

Reply: Indeed, in principle, VCO2 can be turned to $f_r/5$, and the output from the nonlinear mixer is at DC frequency, which can be used to directly feedback control VCO2. In this ideal case, LO2 is not necessary for mmFD.

However, in our experiment, we have to include an LO2 because the DC frequency of the nonlinear mixer needs to be used for mixer biasing, as instructed by the manufacture (Pacific mmWave, model number: WM/MD4A). As a result, close to DC frequency from the mixer cannot be used directly for feedback control. Including LO2 is for a technical reason of our instrumentation, not a limitation of the mmFD method.

We have updated figure 2 in the revised manuscript to show the DC frequency is used for bias. We have also included the model number of the mixer in the main text.

5) On page 3, $f_r = (f_A - f_B - f_{LO1})/N$. Here, f_{LO1} should be Δ .

Reply: We thank the reviewer for pointing this out. The expression in the original manuscript missed a factor of 5 in front of the f_{LO1} . In the servo locking loop, $\Delta/5$ is locked to LO1 by feedback control of the soliton repetition rate, and thus $f_r = (f_A - f_B - \Delta)/N$, or $f_r = (f_A - f_B - 5f_{LO1})/N$.

To clarify this, we modified the sentence in the revised manuscript: “Frequency Δ is *then divided by five electronically and* phase locked to a low-frequency local oscillator (LO, f_{LO1}) by feedback control of the VCO frequency, ... *Within the servo bandwidth*, the frequency and phase of the optical references are thus divided down to the soliton repetition rate, as $f_r = (f_A - f_B - 5f_{LO1})/N$.”

Referee #3 (Remarks to the Author):

In their manuscript on “Integrated optical frequency division for stable microwave and mmWave generation” the authors devise and investigate a technology that allows very good phase noise. The scheme requires commercial ECDL lasers that are stabilized by being locked to a 4 metre long integrated coil cavity. Both of them are then used to stabilize a narrow optical frequency comb with the two point optical frequency division. The resulting phase noise stability is very impressive for an all compact and integratable device. It is still, however, short of the stability reached by fibre-optic/mode lock laser based OFD. In order to quantify the phase noise of the high frequency 100GHz mmWave the authors furthermore develop a new technique that is very interesting.

The work is original and the results significant and well reference, the conclusions are robust.

The achieved power of 9 dBm at 100 GHz is impressive. What would be required to achieve even higher values? What would be required to use this system to go to even higher frequencies up to 300 GHz? Difficult to measure, but powerful light sources would be very sought after.

Reply: The achieved power of 9 dBm is within 1 dB of the highest RF output power that has been reported from a photodiode at 100 GHz to date. To achieve even higher values a photodiode with reduced roll-off in its frequency response as well as improved high-power and saturation characteristics would be needed. Potentially, this can be achieved by downscaling the PD size, a further reduction of the space-charge effect, and improved thermal management by e.g., active cooling.

For the system to go to even higher frequencies up to 300 GHz, a photodiode with ultra-broadband response would be required. We have recently demonstrated photodiodes with detection capability up to 325 GHz, however, their maximum RF output power was -10.5 dBm at 300 GHz [1]. In the meantime, the free-spectral-range (FSR) of the soliton microcomb needs to be increased to the target frequency by reducing the radius of the ring resonator. Soliton repetition rate as high as 1 THz has been shown previously [2].

[1] J. S. Morgan et al., "Bias-Insensitive GaAsSb/InP CC-MUTC Photodiodes for Mmwave Generation up to 325 GHz," in *Journal of Lightwave Technology*, doi: 10.1109/JLT.2023.3298772.

[2] D.T. Spencer, et al., “An optical-frequency synthesizer using integrated photonics,” *Nature* 557 (7703), 81-85 (2018).

Reviewer Reports on the First Revision:

Referees' comments:

Referee #1 (Remarks to the Author):

I am supportive of publication of the revised manuscript.

Referee #2 (Remarks to the Author):

I appreciate the authors for well addressing my comments and I recommend its publications

Referee #3 (Remarks to the Author):

The authors did a fine job responding to my comments and answered the questions of the other referees very well. I have no further comments and think the manuscript is now much enhanced.